# A recurrent neural network framework for flexible and adaptive decision making based on sequence learning

**Zhewei Zhang**[1,2], **Huzi Cheng**[1], **Tianming Yang**[1,3]*

**1** Institute of Neuroscience, Key Laboratory of Primate Neurobiology, CAS Center for Excellence in Brain Science and Intelligence Technology, Chinese Academy of Sciences, China, **2** University of Chinese Academy of Sciences, China, **3** Shanghai Center for Brain Science and Brain-Inspired Intelligence Technology, China

* tyang@ion.ac.cn

**Data Availability Statement:** The source code can be found at: https://github.com/tyangLab/Sequence_learning Other relevant data are within the manuscript.

## Abstract

The brain makes flexible and adaptive responses in a complicated and ever-changing environment for an organism's survival. To achieve this, the brain needs to understand the contingencies between its sensory inputs, actions, and rewards. This is analogous to the statistical inference that has been extensively studied in the natural language processing field, where recent developments of recurrent neural networks have found many successes. We wonder whether these neural networks, the gated recurrent unit (GRU) networks in particular, reflect how the brain solves the contingency problem. Therefore, we build a GRU network framework inspired by the statistical learning approach of NLP and test it with four exemplar behavior tasks previously used in empirical studies. The network models are trained to predict future events based on past events, both comprising sensory, action, and reward events. We show the networks can successfully reproduce animal and human behavior. The networks generalize the training, perform Bayesian inference in novel conditions, and adapt their choices when event contingencies vary. Importantly, units in the network encode task variables and exhibit activity patterns that match previous neurophysiology findings. Our results suggest that the neural network approach based on statistical sequence learning may reflect the brain's computational principle underlying flexible and adaptive behaviors and serve as a useful approach to understand the brain.

## Author summary

The brain faces a continuous stream of sensory events that require variable responses and lead to different reward outcomes. Learning the statistical structure of the event sequence that is composed of sensory, action, and reward events may provide a basis for flexible and adaptive behaviors. This is analogous to how learning text statistics may help natural language processing, where recent developments of recurrent neural networks have found many successes. We construct a neural network framework with a structure similar to those built for natural language processing and test how it works as a model of the brain.

**Funding:** T.Y. was supported by the National Natural Science Foundation of China Project (http://www.nsfc.gov.cn/, Grant No. 31771179), Shanghai Municipal Science and Technology Major Project (http://stcsm.sh.gov.cn/, Grant No. 2018SHZDZX05), and Strategic Priority Research Program of Chinese Academy of Sciences (http://www.cas.cn, Grant No. XDB32070100). The funders had no roles in the study design, data collection and analysis, decision to publish, or preparation of the manuscript.

**Competing interests:** No authors have competing interests

We show that the models based on this framework can reproduce many previous behavioral and neurophysiological findings. These results suggest that statistical sequence learning is a useful approach to understand the brain's decision making and learning.

## Introduction

Consider a scenario in which a cheetah sneaks up on a deer until it jumps out for the final dash to catch it. Every move has to be calculated carefully based on the sensory inputs, for example, the distance to the deer, the surrounding environment, the cheetah's experience and knowledge, and the deer's speed. Furthermore, the cheetah should be able to predict how the deer would respond to its move to allow timely adjustments to its actions. Every action is the consequence of the sensory inputs, and every action, in turn, leads to new sensory inputs. The sensory events, actions, and reward outcomes constitute a stream of events, and the brain needs to learn the contingencies between them for making good decisions.

The general problem of solving contingencies between elements in a sequence interests more than just neuroscientists. In the field of natural language processing (NLP), researchers have been working on similar problems and produced fruitful results [1]. We may draw parallels between the two: an event is analogous to a letter, a string of events that often occur together to a word, a long self-contained event sequence to a sentence, and the statistical structure that describes the contingencies between events to syntax. With these analogies, behavior paradigms ranging from simple Pavlovian and instrumental learning, match-to-sample tasks, reversal learning, to complicated tasks with arbitrary contingency structures, can all be regarded as problems of learning the syntax of the event sequences composed of sensory, action, and reward events instead of letters or words (Fig 1). Although certain sensory-action-reward contingencies may be handled by hard-wired circuitries in the brain, the brain most often has to learn the sensory-action-reward contingencies through noisy experience, which contains a statistical structure. Learning the structure is the key to solving the general contingency problem. With these analogies, we reason that the statistical inference approach that has been used successfully in NLP [1] may also apply to the study of animal's decision making and learning behavior. It may even provide us insights into the underlying brain circuitry.

A category of neural networks has been proven to be especially successful in automatically extracting the statistic features in NLP [2,3]. These networks, including the long-short-term-memory (LSTM) network and the gated recurrent unit (GRU) network, feature a gating mechanism that controls how information is updated in the network units [4–7]. This gating mechanism may be readily achieved with biological neural circuitry composed of excitation and inhibition neurons [8,9] and has been used to model a variety of brain regions and brain functions [10–12]. Therefore, we ask the question of if these networks, combined with the statistical inference approach borrowed from NLP, reflect the neural computation that underlies animals' decision-making and learning behavior.

To answer this question, we build a neural network framework based on the GRU. Analogous to similar networks in the NLP that are trained to predict text sequence, our networks take inputs in the form of event sequences and are trained to predict future events with supervised learning. Importantly, the input event sequences include not only the sensory events but also the action and the reward outcome events. The predictions of action events generate the model's actual responses, which are assessed for the network's behavior performance. With four exemplary behavior tasks, we show that the network models are able to learn tasks with extended and variable length and make choices based on the contingencies between events far

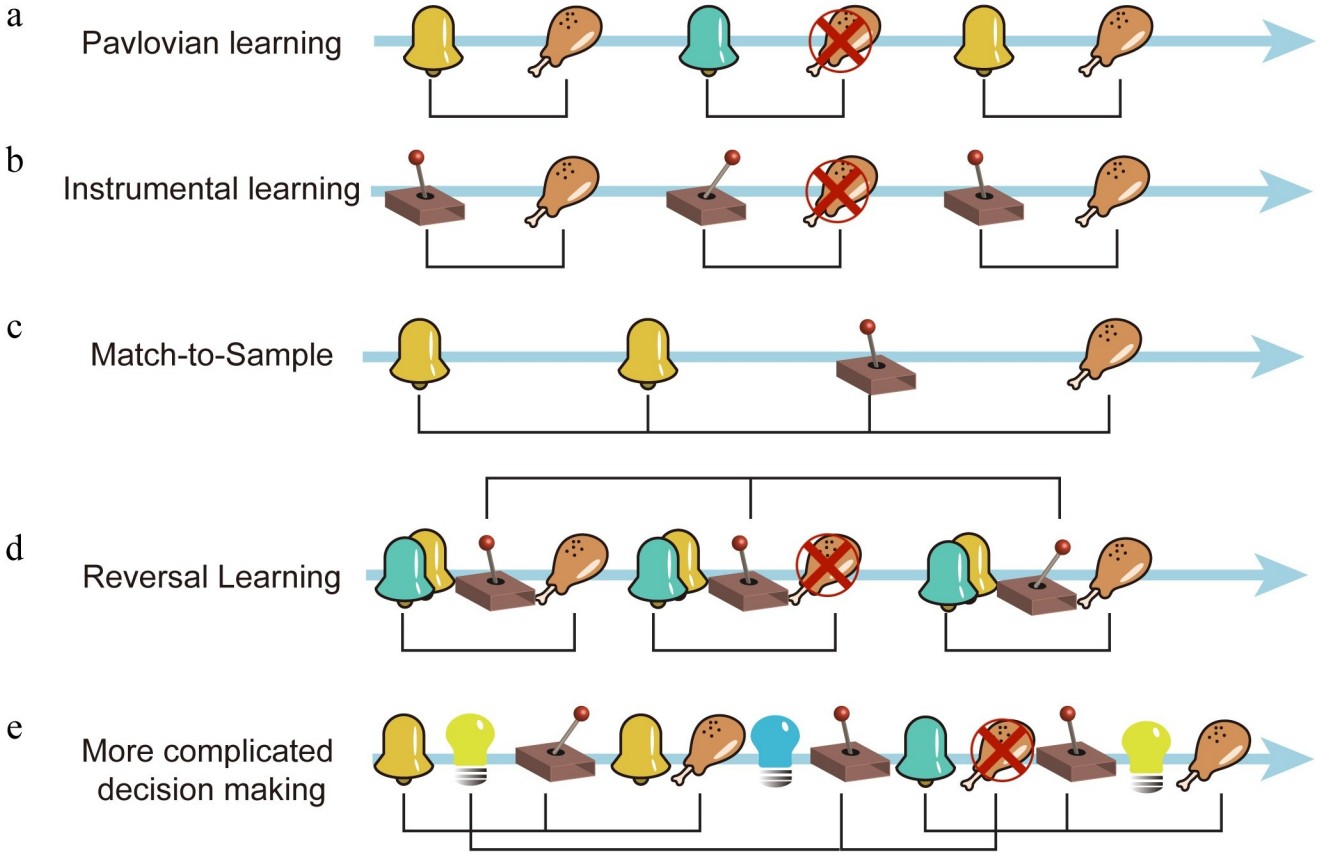

**Fig 1. Sequence learning and decision making. a.** Pavlovian learning. Different cues predict different reward outcomes. **b.** Instrumental learning. Different actions lead to different reward outcomes. **c.** An example match-to-sample task. Moving the lever leftward after a pair of matching cues leads to a reward. **d.** Reversal learning. Two options are presented. The left choice is initially rewarded, but the reward switches to the rightward choice in the second trial. Notice both the contingencies between events within each trial and events across trials are essential for the learning. **e.** In more complicated decision making, the contingencies can be between many different types of sensory, action, and reward events distributed across time. Black brackets indicate contingencies that exist.

apart from each, including those across trials. The learning can be generalized to novel sequences and even novel conditions. Importantly, we show that the activity patterns of the network units resemble those of the neurons in the brain, which reflect accumulated evidence, choice, and value. These results, along with the similarities between our framework and the neural circuitries of decision making in the brain, suggest that the computational principle in the brain may be understood as a sequence learning process based on statistical inference.

## Results

### Network framework

Our framework contains three layers: the input layer, the hidden layer based on the gated recurrent units (GRU), and the output layer (Fig 2A). The input layer contains units that carry the information about the sensory, motor, and reward events on a timeline. Each unit's activity can be a binary variable, representing the presence or the absence of the corresponding event, or a continuous variable representing stimulus strength. The input units are fully connected with the next layer. We use vector $x_t$ of length $N_{IL}$ to describe the activities of the input layer units.

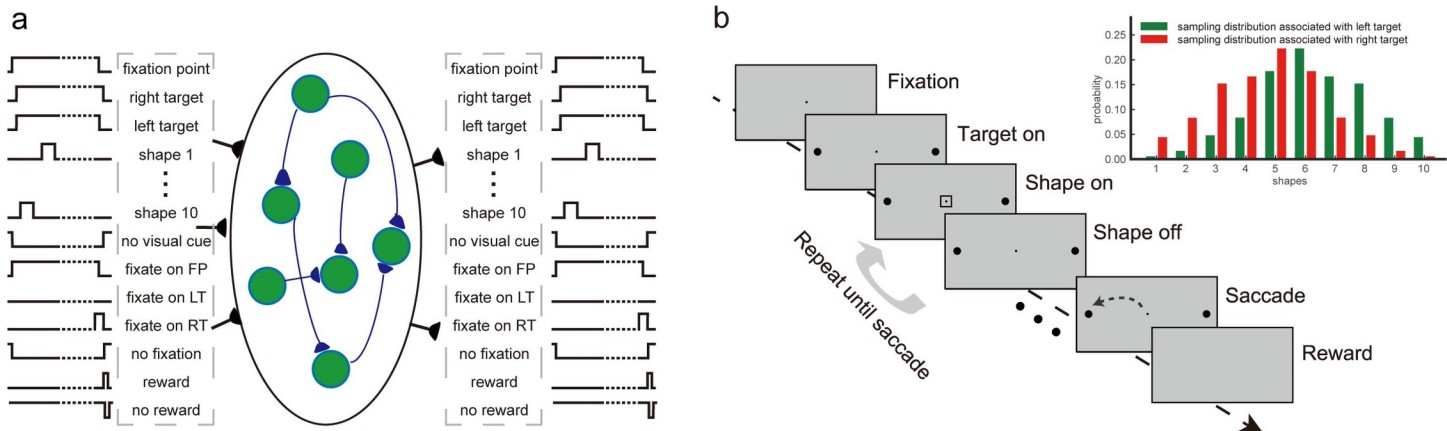

**Fig 2. The network framework and Task 1: Probabilistic reasoning. a.** The framework diagram. The network has three layers: the input layer, the hidden layer, and the output layer. The input layer receives the input sequences of sensory, action, and reward events. The hidden layer has 128 gated recurrent units. The output layer units mirror the input layer units and represent the prediction of future events. The diagram illustrates the particular input and output units for Task 1. **b.** Task 1: the reaction-time version of probabilistic reasoning task. The subject fixates at a central point and views a series of shapes to make a response by moving the eyes toward one of the two choice targets on the peripheral. Each shape confers information regarding which target will be rewarded. The optimal strategy is to integrate the information and to make a choice when the integrated information hits a bound. The inset shows the sampling distributions.

The hidden layer is the core of our framework, which is a GRU network. There are $N_h =$ 128 units in the hidden layer. Each gated recurrent unit's activity is a nonlinear combination of both the inputs and the units' activities at the previous time step, which are regulated by its update gate and reset gate. The state vector $h_t$ of length $N_h$ describes the responses of these units at time $t$, which is updated as follows:

$$z_t = \text{Sigmoid}(W_z x_t + U_z h_{t-1} + b_z), \tag{1}$$

$$r_t = \text{Sigmoid}(W_r x_t + U_r h_{t-1} + b_r), \tag{2}$$

$$s_t = (1 - z_t) \bullet h_{t-1} + z_t \bullet \tanh(W_h x_t + U_h(r_t \bullet h_{t-1}) + b_h), \tag{3}$$

$$h_t = [s_t]_+, \tag{4}$$

where $z_t$ and $r_t$ represent the update gate vector and the reset gate vector, respectively; $W_z$, $W_r$ and $W_h$ are the input connection weight matrices for the update gates, the reset gates, and the gated units; $U_z$, $U_r$ and $U_h$ are the recurrent connection weight matrices for the update gates, the reset gates, and the gated units; $b_z$, $b_r$ and $b_h$ are the bias vectors for the update gates, the reset gates, and the gated units; and $\bullet$ indicates the element-wise multiplication. Unlike standard GRU networks, we use a rectified-linear function $[\bullet]_+$ to keep $h_t$ non-negative. The initial value of $h_t$ at the beginning of each trial is reset to zero.

The hidden layer units project to the output layer with full connections. The output layer is composed of units that mirror the input layer units, representing the network's predictions of the corresponding sensory, action, and reward events for the next time step. The activity of the output layer units $y_t$ (a vector of length $N_{OL}$) is:

$$y_t = \text{Sigmoid}(W_o h_t), \tag{5}$$

where $W_o$ is the output connection weight matrix.

The prediction of the sensory and reward inputs, $o_t$, is a function of response $y_t$:

$$o_t = \begin{cases} 1 & y_t > 0.5 \\ 0 & y_t < 0.5 \end{cases}. \tag{6}$$

In the current model, only one action can be executed at one time. The probability of action $a_t^i$ being executed is a softmax function of $y_t$:

$$P(a_t^i | \textstyle\sum_{j=1} y_t^j) = exp(\beta * y_t^i) / \textstyle\sum_{j=1} exp(\beta * y_t^j), \tag{7}$$

where $\beta$ is the inverse temperature, $a_t^i$ stands for the $i$-th action at time $t$, $y_t^j$ are the activities of action output unit $\underline{i}$ at time $t$. The chosen action is evaluated as the network's performance during the testing phase.

Fig 2A illustrates the network structure with the inputs and outputs corresponding to Task 1 that we discuss below. For the other tasks, we only change the input and output units according to each task's requirements.

The goal of the training is for the network to predict events and generate sequences that lead to a reward. The loss function $L$ is defined as the sum of mean squared error between elements in the output $y_t$ and actual event sequence $x_{t+1}$ for all time points $t$.

$$L = \textstyle\sum_{t=1}^{T-1} (y_t - x_{t+1})^2 / N_{OL} \tag{8}$$

where $T$ is the length of each trial. The parameter vector $\theta_k$ (including $W_*$, $U_*$ and $b_*$, $k$ indicates each individual parameter) of the network is updated with Adam [13] in the end of each trial:

$$\theta_k^t = \theta_k^{t-1} - \eta \frac{m_k^t}{\sqrt{v_k^t} + \epsilon}, \tag{9}$$

$$m_k^t = \beta_1 m_k^{t-1} + (1 - \beta_1) g_k^t, \tag{10}$$

$$v_k^t = \beta_2 v_k^{t-1} + (1 - \beta_2) [g_k^t]^2, \tag{11}$$

where $\eta$ is the learning rate, $m_k^t$ and $v_k^t$ denote the first and second moments of the gradient, respectively. At the beginning of the training, $m_k^0$ and $v_k^0$ are set to zero. $\beta_1 = 0.9$, $\beta_2 = 0.999$, $\epsilon = 10^{-8}$, $\eta = 10^{-3}$ in the first three tasks and $\eta = 10^{-4}$ in the two-step task. In addition, a gradient clipping is applied to avoid exploding gradients as follows:

$$g_t = \frac{\partial L}{\partial \theta} \big|_{\theta = \theta_{t-1}} * min \left( 1, \frac{0.25}{|\frac{\partial L}{\partial \theta}|_{\theta = \theta_{t-1}}|} \right) \tag{12}$$

We test the framework with four exemplar tasks that are aimed at different aspects of decision making and learning behavior. The models trained to do these tasks share the same structure, with the exception that the inputs and outputs layers reflecting the requirement of each task. The results presented here are based on 20 independent runs for each task. The model in each run is trained with a dataset containing $7.5 \times 10^5$ trials.

## Task 1: Probabilistic reasoning task

We train the first model with a reaction-time version of the probabilistic reasoning task that was used to study the neural mechanism of sequential decision making [14]. The task is

illustrated in Fig 2B. In this task, a subject has to make decisions between a pair of eye movement targets. Initially, the subject needs to fixate on a central point on a computer screen. Then a stream of shapes appears sequentially near the fixation point. There are 10 possible shapes. Each shape conveys information on the correct answer. The subject needs to integrate information from the shapes to form a decision and choose the corresponding choice target whenever ready.

Two distributions describe the contingency between the shapes and the targets' reward. Each target is associated with a distribution of the appearance probability of the shapes in the sequence (Fig 2B inset). In each trial, the computer randomly picks the correct target. Shapes are independently sampled from the distribution associated with the correct target. Because the likelihoods of observing a particular shape under the two distributions are different, each shape provides information on which target is correct. It has been shown that the sequential probability ratio test (SPRT) is an optimal strategy to solve the task in the sense that it requires the least number of observations to achieve a desired performance [15]. In the SPRT, one needs to accumulate the log likelihood ratio (logLR) associated with each shape, which is the log ratio between the conditional probabilities of observing the shape given the two testing hypotheses. In our task, the logLRs associated with each shape range from -0.9 to 0.9 (base 10). We define positive and negative logLRs as evidence supporting the left and the right target, respectively.

The task has several attractive features for our modeling purposes. It features a sophisticated statistical structure describing how the shapes, the choice, and the reward are related. As the shapes appear one by one, an ideal observer not only gains information on what the appropriate choice would be but also can deduce how likely a particular shape will appear next and a reward can be expected. In addition, The reaction-time aspect of the task allows the choices to be made at any time and lets us test the flexibility of our model for learning sequences of variable length. Last but not least, the task is one of the most complicated tasks that have been used in animal studies with both behavior and neuronal data available. We not only can compare the behavior between the models and the animals but also can look into the network and study the network units' activities and compare them against the experimental findings.

**The training dataset.**   We train the network with task event sequences created with simulated trials in which choices are generated with a drift-diffusion model (DDM) with collapsing bounds. In each trial, a correct target is randomly determined, and the shapes are generated with its associated distribution (Fig 2B inset). A choice is triggered when the accumulated logLR reaches either of the two opposite collapsing bounds. The bounds start at ±1.5 and linearly collapse toward 0 at the rate of 0.1 per shape epoch. The left target is chosen if the positive bound is reached, and the right target chosen if the negative bound reached. If the choice matches the pre-determined correct target, a reward is given. Only the events in the rewarded trials are used in training, but the results hold if all trials are used, since most choices generated with the DDM lead to rewards.

The results presented below are based on 20 simulation runs using a training set generated with $7.5 \times 10^5$ simulated trial sequences.

**Performance.**   During the testing sessions, the network is fed with randomly generated shape sequences, but the eye movement decisions, whether to hold the fixation or to commit to a choice by saccading to the corresponding target, are now generated at each time step by the network with a softmax function based on the activity of the output units corresponding to fixation, leftward saccade, and rightward saccade (Eq 7). There are no additional mechanisms or post-processing stages. Based on the choices, subsequent sensory and reward inputs are given. We evaluate the network's choice accuracy and its reaction times.

After training, the network performs the task well. When the total logLR associated with the shape sequence is positive, the network tends to choose the left target, and when the total

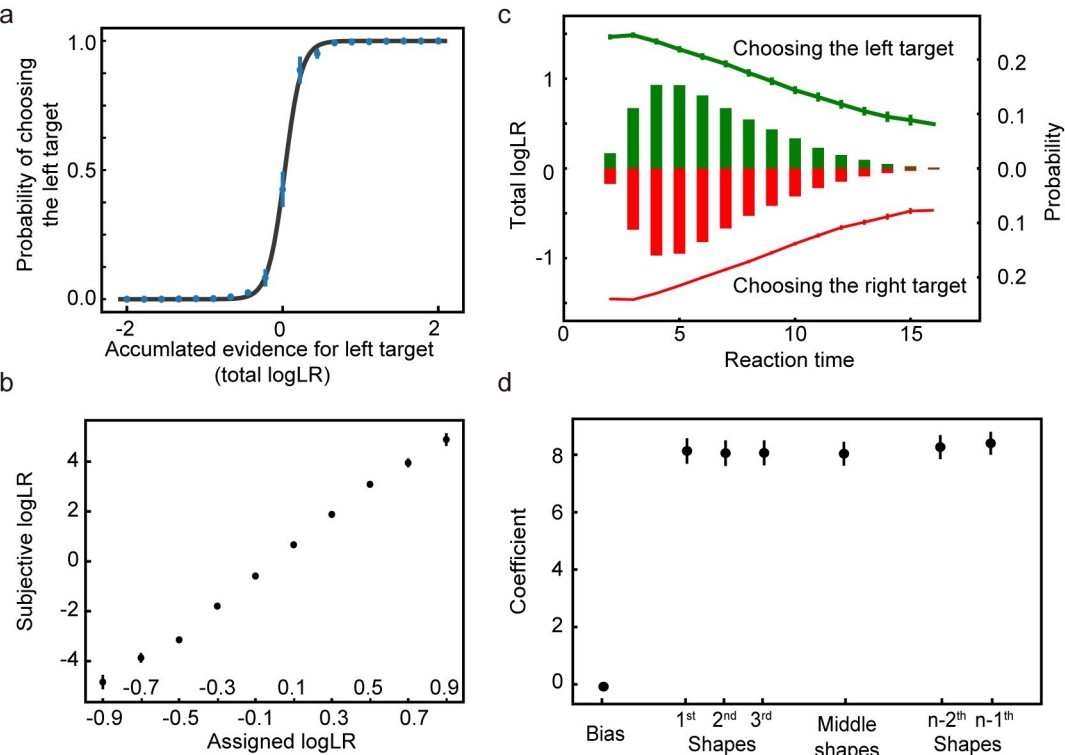

**Fig 3. Task 1: Behavioral analyses. a.** The psychometric curve. The model more often chooses the target supported by the accumulated evidence. The black curve is the fitting curve from the logistic regression. **b.** The leverage of each shape on choice revealed by the logistic regression is consistent with its assigned logLR. **c.** Reaction time. The bars show the distribution of reaction time, quantified by the number of observed shapes (right y-axis). Green and red indicate the left and right choices, respectively. The lines indicate the mean total logLR (left y-axis) at the decision time, grouped by reaction time. Trials with only 1 shape or more than 16 shapes comprise less than 0.1% of the total trials and are excluded from the plot. **d.** The leverage of the first 3, the second and third from the last, and the middle shapes on the choice. Only trials with more than 6 shapes are included in the analysis. No significant differences are found between any pair of the coefficients of shape regressors (two-tailed t-test with Bonferroni correction). The error bars in all panels indicate SE across runs. Some error bars are smaller than the data points and not visible.

logLR is negative, it tends to choose the right target (Fig 3A). A logistic regression model reveals that the logLR assigned to each shape correlates well with its leverage on the choice (Fig 3B). Both the reaction time, quantified as the number of shapes used for decisions, and the total logLR at the time of choice can be well described with a DDM with collapsing bounds (Fig 3C, 94.00 ± 0.60% variance captured). We use another logistic regression model to examine the effects of shape order on the choice. The first shapes in the sequence exert similar leverages on the choice to the rest of the shapes, except for the last shape, suggesting all shapes except the last are used equally in decision making (Fig 3D). The last shape's sign is almost always consistent with the choice (> 99% of all trials). This is consistent with the DDM, in which the last shape takes the total logLR over the bound. Overall, the network performance resembles, although understandably better than, the behavior of the macaque monkeys trained with a very similar task [14].

The good performance of the network is not because the trial sequences in the testing dataset overlap with those in the training dataset significantly. Each shape sequence in the training and testing dataset is randomly generated. The possible number of trial sequences is astronomical. Even when the network model is trained with a dataset of the same number of trials but containing only 1000 unique trial sequences, it still achieves good performance using the same

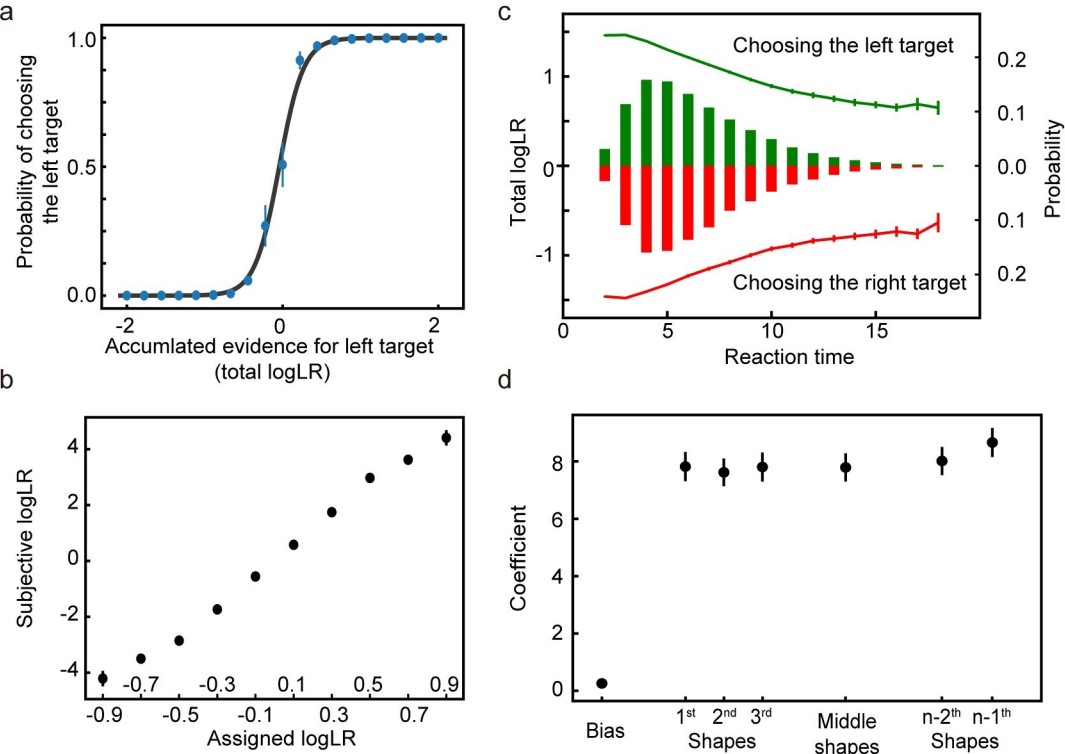

**Fig 4. Task 1: Model training with limited dataset.** Same conventions as in Fig 3. The training dataset contains only 1000 unique sequences. **a**. The psychometric curve. **b**. The leverage of each shape on choice. **c**. Reaction time distribution (bars, right y-axis) and the mean total logLR (lines, left y-axis) at the decision time. Green and red indicate the left and right choices, respectively. **d**. The leverage of the first 3, the second and third from the last, and the middle shapes on the choice. The error bars in all panels indicate SE across runs. Some error bars are smaller than the data points and not visible.

testing dataset as before (Fig 4). Therefore, the network is able to generalize beyond the training dataset, and the learning of the statistical structure of the task is likely the reason.

**Network analyses–evidence and choice encoding.** Next, we examine how the evidence and the choice are encoded in the network. The example unit in Fig 5A shows a classical ramping-up activity pattern that has been reported in neurons from the prefrontal cortex [16], the parietal cortex [14,17,18], and the striatum [19]. Its activity increases when the total logLR grows to support its preferred target and decreases when the total logLR is against its preferred target. The responses converge around the time when its preferred target is chosen. The population analysis using all the units that are selective to the total logLR during the delay period finds the same pattern (Fig 5B).

In addition to the units that accumulate evidence and reflect the decision-making process, we also find units that show a ramping-up or ramping-down activity pattern independent of the choice (Fig 5C). Their activities indicate the passage of time and may be interpreted as an urgency signal. Neurons with a similar activity pattern have been reported in the global pallidus [20].

We use linear regression to quantify the selectivity of each unit (see Methods) to four critical parameters for this task: the accumulated evidence, quantified with the total logLR, the choice outcome, the urgency, and the absolute value of the accumulated evidence. During the shape presentation period, a large proportion of units encode the accumulated evidence (N = 63.00 ± 2.47), the choice outcome (N = 50.60 ± 2.03), the urgency (N = 86.50 ± 1.10), and the absolute value of the accumulated evidence (N = 92.8 ± 1.13).

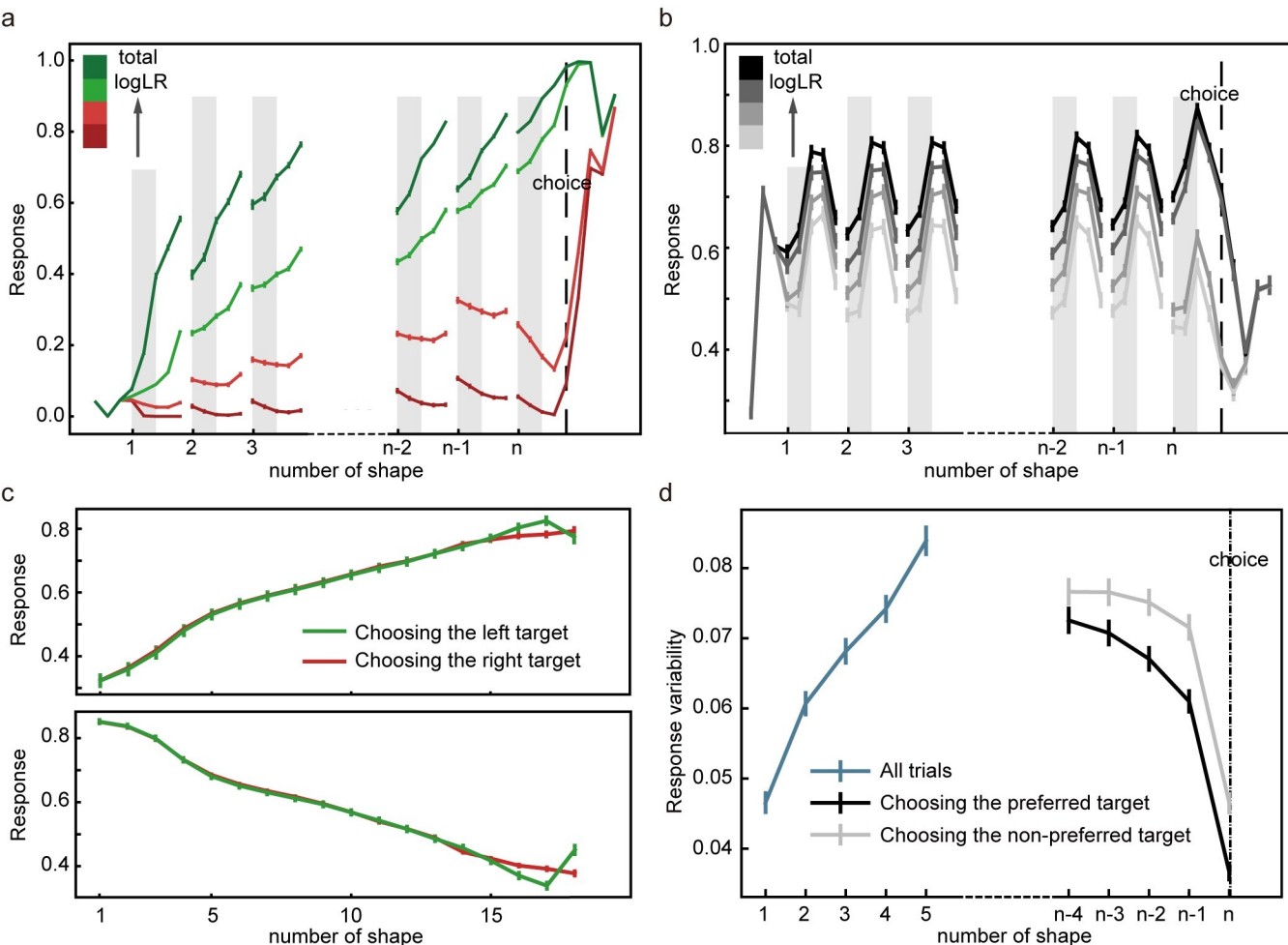

**Fig 5. Task 1: Network units' responses. a**. An example unit that prefers the left target. Its activity increases when the evidence supporting the left target grows and decreases when it drops. The unit's responses converge when the network chooses its preferred target. The trials are grouped into quartiles by the total logLR in each epoch. The colors indicate the quartiles, and the error bars indicate the SE across trials. **b**. Population responses of the units that are selective to the total logLR. The trials are grouped based on the total logLR supporting each unit's preferred target in each epoch. The error bars in panels **b**, **c**, and **d** indicate the SE across units. **c**. Urgency units. Their activities ramp up (upper panel) or down (lower panel) regardless of choice. **d**. Network unit response variability. The neurons' response variability increases initially (blue curve) but decreases before the choice, more so when the preferred target is chosen (black) than when the non-preferred target is chosen (grey). Only the trials with more than five shapes are included in panel **a**, **b**, and **d**.

Finally, neuronal response variability has been measured experimentally for the investigation of neural computation during decision making [21], which provides us a nice opportunity to compare our model against the experimental findings. Analogous to the variance of conditional expectation (VarCE) that Churchland and colleagues studied previously in the LIP neurons [21], we calculate the response variability of the units in the hidden layer (Fig 5D). Because the units in our model do not have intrinsic Poisson noise, the response variability in our model is calculated as the standard deviation across trials of the responses of the units that are sensitive to the accumulated evidence. We find that the response variability increases initially (linear regression, k = 0.0088, p < 0.001) as more shapes are presented and evidence is accumulated. Furthermore, a different pattern emerges when we align the trials to the choice. In the trials in which the preferred target is chosen, the response variability decreases before the choice (linear regression, k = -0.0038, p < 0.001; Spearman's rank correlation, r = -0.045, p < 0.01) and reaches the minimum around the time of choice. In contrast, in the trials in

                                    

which the non-preferred target is chosen, the response variability is significantly higher. Its decrease is not significant before the last shape (linear regression, k = -0.0017, p = 0.07; Spearman's rank correlation, r = -0.017, p = 0.28). The overall pattern reflects the evidence-accumulation process during decision making and is similar to that of the LIP neurons reported previously [21].

**Network analyses–when and which.** The balance between speed and accuracy is an important aspect of decision making. However, it is unknown whether the speed-accuracy tradeoff is regulated through the same neurons that determine the choice [22–24] or through a distinct population of neurons that reflect only the speed but not the choice [20]. Here, we analyze the units' activities and connectivities in the model to find out which is the case in our model.

We first identify the units in the hidden layer that may contribute to the network's reaction time and choice. We examine the connection weights between hidden layer units and the output units related to the decision. The hidden layer units that are differentially connected to the left (LT) and the right (RT) saccade units most likely contribute to the left-versus-right choice, and those that are differentially connected to the fixation (FP) unit and the two saccade units may contribute to the hold-versus-go decision, which determines the reaction time. Therefore, we define two indices for each hidden layer unit, $I_{when}$ and $I_{which}$, to quantify its contributions to two kinds of decisions:

$$I_{when} = (w^i_{ho-LT} + w^i_{ho-RT})/2 - w^i_{ho-FP} \tag{13}$$

$$I_{which} = w^i_{ho-LT} - w^i_{ho-RT}, \tag{14}$$

where $w^i_{ho-LT}$, $w^i_{ho-RT}$ and $w^i_{ho-FP}$ are the connection weights between the hidden layer unit $i$ and the output layer units LT, RT, and FP, respectively. Units with positive $I_{when}$ promote the saccades over fixation and shorten the reaction time. In contrast, units with positive $I_{which}$ bias the saccade direction toward the left.

Next, we divide units into different groups based on their indices and examine how manipulating each group affects the network behavior. We define the +*when* group units with the following procedure. First, we sort all units with positive $I_{when}$ values by their $I_{when}$ values in the descending order. Then, we calculate the accumulative $I_{when}$ along this axis and select the top units that together contribute more than 50% of the sum of the positive $I_{when}$ as the +*when* units (10.70 ± 0.26 out of 50.60 ± 0.89 units). They have larger connection weights to the two saccade output units than to the fixation output unit (Fig 6A). We use similar procedures to select the -*when* group units, which contribute to the reaction time antagonistically (n = 13.80 ± 0.27 out of 77.40 ± 0.89 units). We also use $I_{which}$ to define the +*which* and -*which* group units (+*which*: 9.80 ± 0.30, -*which*: 9.50 ± 0.28; out of 64.00 ± 1.25 units), which may contribute most to the left-versus-right choices. The *when* and *which* group units only overlap rarely (N = 1.80 ± 0.35), suggesting that two distinct sub-networks contribute to the reaction time and the choice separately. All four groups of units constitute 34.22 ± 0.52% of all the units in the hidden layer. The connection weights of the both *when* groups units are biased toward the positive side for the fixation and the negative side for the left/right saccade (Fig 6A), because the network keeps fixation most of the time. Similarly, the average connection weights between the *which* units and the left/right saccade units are also negative (Fig 6A), suggesting that the *which* units are also slightly biased toward holding the fixation.

We confirm the roles of each group by manipulating each group of units' output connections. We selectively sever the output connections of a selected group of units, leaving the hidden layer itself intact. When we remove the +*when* units' outputs, the RTs become longer

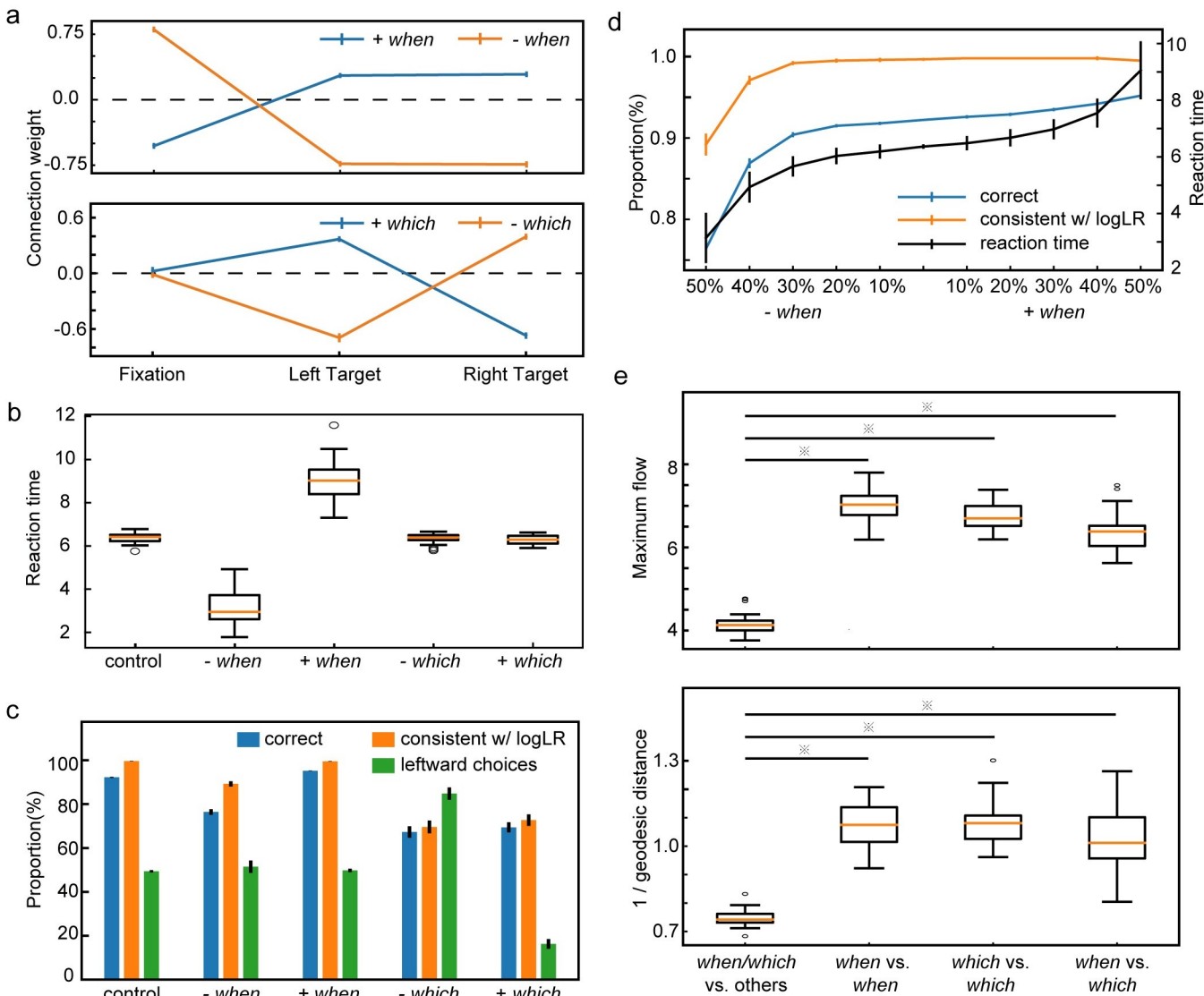

**Fig 6. Task 1: *Which* and *when* units. a**. The connection weights between the eye movement output units and the *when* units (upper panel) and the *which* units (lower panel). **b**. The reaction time of the network choice when the outputs of different groups of units are inactivated. **c**. Lesions to the *when* and *which* units affect choices differently. The blue bars indicate the proportion of correct trials. The orange bars indicate the proportion of trials in which the choice is consistent with the sign of the accumulated evidence at the time of choice. The green bars indicate the percentage of trials in which the model chooses the left target. **d**. Speed-accuracy tradeoff. We suppress the output of a different proportion of +*when*/-*when* units (see Methods). As more +*when* units' outputs are suppressed, the model's reaction time (black curve, right y-axis) increases along with the accuracy (blue curve, left y-axis). However, the proportion of trials in which the choices are consistent with the evidence (orange curve, left y-axis) stays the same except for the extreme cases. **e**. The maximum flow (upper panel) and the inverse of the geodesic distance (lower panel) between different unit groups. The smaller maximum flow and the larger geodesic distance between *when*/*which* units and other units suggest the relatively tight connections between the *when* and *which* units. ※ indicates a significant difference (p<0.05, Two-tailed t-test with Bonferroni correction). The error bars in all panels indicate the SE across runs.

while the accuracy remains intact (Fig 6B and 6C). In contrast, when we remove the outputs of the -*when* units, the network's RTs become shorter (Fig 6B). Although the network performance has decreased by 14.3%, the network's choices are still mostly consistent with the accumulated logLR (89.16 ± 1.23%, a decrease of 9.5%), suggesting that the performance drop reflects the fact that the network has to work with a smaller total logLR due to the shorter RT (Fig 6C). In comparison, when *which* units are manipulated, only the choice accuracy is affected, but the RT remains the same (Fig 6B and 6C). More specifically, inactivating +*which*

units leads to a bias toward the right target, while inactivating -*which* units leads to a bias toward the left target (Fig 6C). These results suggest there are two distinct populations of hidden layer units contributing to the choice and the reaction time separately.

The way how the *when* units affect the accuracy and reaction time suggests a possible mechanism to modulate the speed-accuracy tradeoff. We demonstrate this by suppressing different amounts of *when* units' outputs (see Methods for details). As more +*when* units' outputs are suppressed, the reaction time becomes longer (Spearman's rank correlation, p<0.001), and the accuracy becomes higher (Spearman's rank correlation, p<0.001) (Fig 6D). Importantly, the choices remain consistent with the accumulated evidence except for the most extreme cases.

The distinct functions of the *when* and *which* units may suggest they are also topologically separated. To test this hypothesis, we construct a weighted directed graph with the connection matrix of the hidden layer units and calculate the geodesic distance and the maximum flow between the hidden layer units (see Methods). To account for the units that are not connected, we use the inverse of the geodesic distance. It is defined as 0 between disconnected units. Interestingly, the average geodesic distance between the *when* units and the *which* units is significantly smaller than that between the *when*/*which* units and the others. The average maximum flow between the *when* units and the *which* units is also significantly higher than the network average (Fig 6E). These results suggest that the *when* and the *which* units belong to a tightly connected sub-network within the hidden layer and are not topologically separated.

**Network analyses–predictions of sensory inputs.** So far, we have shown that our network can make appropriate choices based on shape sequences. In addition, as one accumulates information regarding the correct target, the distribution from which the upcoming shapes are drawn can also be inferred. Because the network is trained to predict the sensory events as well, the output units related to the shapes should provide predictions of future shapes.

We plot the mean subthreshold activities $y_t$, of the ten shape output units in Fig 7A at the time point right before the onset of each shape. When the current evidence is in favor of the left target, the activities of the shape output units resemble the probability distribution associated with the left target. The units tend to encode the right target's distribution when the evidence is in favor of the right target (Fig 7A).

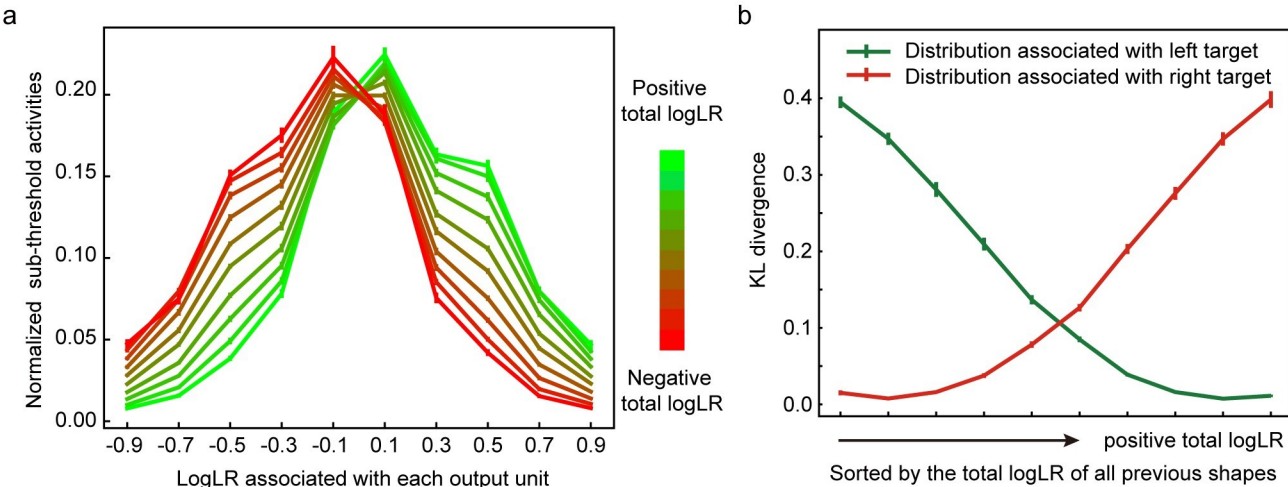

**Fig 7. Task 1: Sensory predictions. a**. The normalized subthreshold activities of 10 shape output units. We show the shape output units' activities at the time step immediately before each shape onset for all epochs in all trials. The sum of the activities of all shape output units is normalized to 1. Data are divided into 10 groups by the total logLR before the shape onset, which is indicated by the color. **b**. The Kullback-Leibler (KL) divergence between the normalized subthreshold activities (as shown in Fig 7A) and the sampling distributions (shown in Fig 2B inset). Data are grouped by the total logLR. The error bars indicate the SE across runs.

We quantify the similarity between the activity profile of the output layer shape units and the sampling distributions by calculating the Kullback-Leibler (KL) divergence (Fig 7B). The KL divergence between the output layer unit activities and the sampling distribution associated with the left target decreases as the total logLR supporting the left target gets larger. It grows when the total logLR is smaller. The opposite trend is observed in the KL divergence between the output layer unit activities and the sampling distribution associated with the right target. These results suggest that the activities of the output layer shape units encode the probability distribution of the upcoming shape based on the accumulated evidence.

## Task 2: Multisensory integration task

The decision-making in Task 1 may also be interpreted as a Bayesian inference process in which each shape updates the posterior reward probability of each target. Next, we test whether the framework can also model how the brain combines uncertain information from different sources. Task 2 is a multisensory integration task adapted from Gu et al., 2008 [25]. In this task, subjects have to discriminate the heading direction, left or right, based on either the visual input, the vestibular input, or both. Gu and colleagues showed that the animals were able to integrate information from different modalities optimally and reached a behavioral threshold that matched predictions based on Bayesian inference.

We train a model with our network framework (See Methods). Critically, the training is limited to the unimodal conditions in which inputs are from only one single sensory modality, and the testing dataset includes both the unimodal and the bimodal conditions. We find that the network can generalize the training to the bimodal condition. With the information available from both the visual and vestibular inputs, the network achieves higher accuracy without further training (Fig 8A). We fit the network model's choices in both the unimodal and bimodal conditions with cumulative Gaussian functions and define the performance threshold as the standard deviation of the best-fitting function. The threshold under the bimodal condition is significantly smaller than the thresholds under the unimodal conditions (Fig 8B). Importantly, it matches the prediction of Bayesian inference. These results suggest that the network model is able to perform Bayesian inference without being explicitly trained to do so.

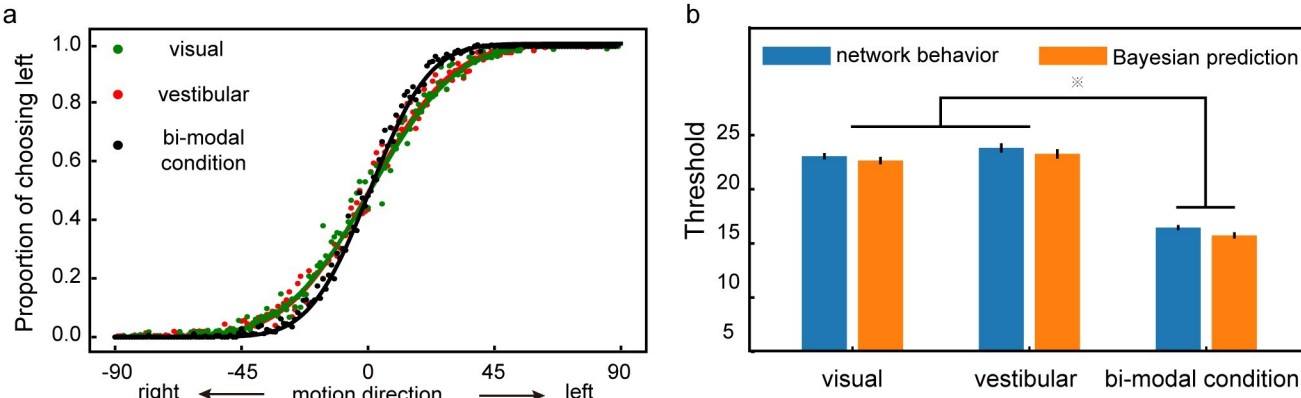

**Fig 8. Task 2: Multisensory integration. a**. The psychometric curve. The model is trained with the unimodal conditions and tested with both the unimodal (green: visual, orange: vestibular) and the bimodal (black) conditions. Each data point represents the proportion of the left choice for a given motion direction. The model performs better and shows a steeper psychometric curve for the bimodal condition. **b**. The performance thresholds. The model's thresholds (blue) are compared against the thresholds calculated with the optimal Bayesian inference (orange). The thresholds under the bimodal condition are significantly lower than those under either unimodal condition. The differences between the thresholds of the network and the thresholds calculated with Bayesian inference are not significant. (two-tailed t-test with Bonferroni correction, p-value threshold = 0.05) The error bars indicate the SE across runs.

## Task 3: Confidence / post-decision wagering task

It has been argued that the same neurons that represent the decision variable during decision making may also support meta-cognition, such as confidence. Previously, a post-decision wagering task was used to test this hypothesis experimentally [26]. In this task, the subject needs to make a decision about the movement direction of a random dot motion stimulus. The task difficulty is controlled by the proportion of dots moving coherently and the duration of the random dot stimulus. A reward is delivered if the decision matches the direction of the coherently moving dots. In half of the trials, a third target (sure target) appears after the motion viewing period. Choosing the sure target leads to a smaller but certain reward. The subject may opt to choose the sure target when its confidence about its decision on the dots' movement direction is low. The confidence, which is based on the assessment of the quality of the decision, is a form of meta-cognition.

To find out whether our framework may support meta-cognition, we train a model with the same task and test its performance. The trained network exhibits a similar choice pattern to the monkeys' behavior [26]. When the motion strength is weaker and the stimulus duration is shorter, the task difficulty is greater. As a result, the network chooses the sure target more frequently (Fig 9A). In addition, because the model may choose the sure target when it is available and the network's confidence about the motion direction is low, the overall accuracy becomes higher in trials with the sure target than without (Fig 9B). The model's behavior is consistent with that of the monkeys [26].

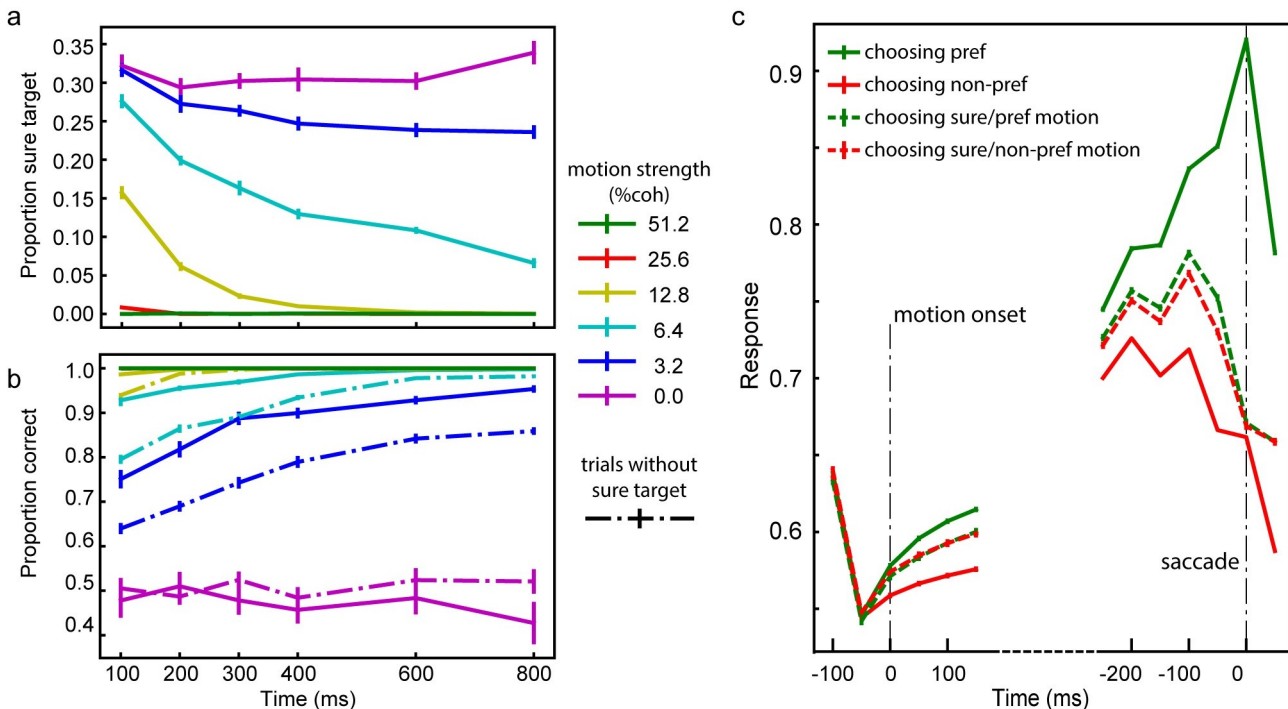

**Fig 9. Task 3: Post-decision wagering. a**. The proportion of trials in which the model chooses the sure target. The color indicates the motion strength. The frequency of the sure target choices decreases with the motion strength and the duration of motion viewing. **b**. The accuracy. Solid lines are the trials with the sure target, and dashed lines are the trials without the sure target. The accuracy is higher when the sure target is available but not chosen. The error bars in panels **a** and **b** are SE across runs. **c**. Activities of choice-selective units. The responses are aligned to the movie onset (left) and the choice (right). The color of each line denotes the choice and motion direction. The dashed lines are trials in which the model chooses the sure targets. The units have an intermediate activity level in trials that the sure target is chosen. The error bars are SE across units with choice selectivity in all runs.

We further look at the response patterns of the units in the hidden layer. In particular, we want to study the units (N = 89.2±1.20 out of 128) that show choice selectivity during the delay period before the choice. This selection criterium is similar to what was employed in the experimental study (Kiani and Shadlen, 2009, see Methods). The responses of these units start to diverge according to stimulus strength during the motion viewing period (Fig 9C). More importantly, in the trials in which the network chooses the sure target, their responses reach an intermediate level between the activity levels when the preferred and the non-preferred targets are chosen. These units behave like the neurons recorded from the LIP area and support the post-decision wagering [26].

## Task 4: Two-step task

In the last section, we extend our investigation to a task in which the reward contingencies change over time. For the framework to exhibit learning, it has to incorporate trial history information for future decisions. In other words, it needs to learn the contingencies across trials. By piecing events across multiple trials together into an extended sequence, we train our network model to learn to adapt its choices based on past trials.

Task 4 is adapted from the original two-step task that was used to study model-based reinforcement learning behavior in humans [27,28]. In the task, the agent has to choose between two options A1 and A2, each leading to one of the two intermediate outcomes, B1 and B2, with different but fixed transition probabilities (Fig 10A). In particular, A1 is more likely followed by B1 (common) than B2 (rare), and A2 is more likely to be followed by B2 (common) than B1 (rare). Finally, B1 and B2 are each associated with a probabilistic reward (0.8 vs. 0.2). The reward contingencies of B1 and B2 are reversed across blocks. Each block consists of 50 trials. Once the agent understands the task structure, it will adapt its choices based on the intermediate states B1 and B2 instead of the initial choice between A1 and A2.

The training dataset consists of trials with randomly assigned choices, which aims to reflect that the actual learning by human subjects starts from rather random choices. During the training, only the events in the rewarded trials are actually trained. This is because the goal of the training is to learn the sequences that are more likely to lead to a reward. If we include both rewarded and unrewarded trials during training, the network would learn choices that lead to both rewards and no rewards. Although the reward outcome of a particular action is predictable, there are no a mechanism in the current model to choose between the actions based on this prediction of rewards, and the network would only generate random choices.

For the network to learn the event contingencies across trials, the state vector $h_t$ is not re-initialized at the beginning of each trial. The loss function is calculated based only on the current rewarded trial, but the errors are back-propagated to the previous trial. The network connections are updated accordingly. In the testing sessions, the network connections are no longer updated.

After the training, the network learns to adjust its choices when the reward contingencies switch. Immediately after a block change, the network performs poorly because of the reversed reward contingency. Its performance, then, recovers gradually (Fig 10B). The network does not merely learn to reverse its choice at a fixed number of trials. We train the network with 50-trial blocks and test it with a different block size of 70 trials. Similar results are observed (Fig 11).

To further investigate the network's adaptive behavior, we use a factorial analysis introduced by Daw and colleagues [27] to look at how different factors of the task affect the choices. We sort the trials based on the intermediate outcomes and the reward outcomes into four groups: common-rewarded (CR), common-unrewarded (CU), rare-rewarded (RR), and rare-

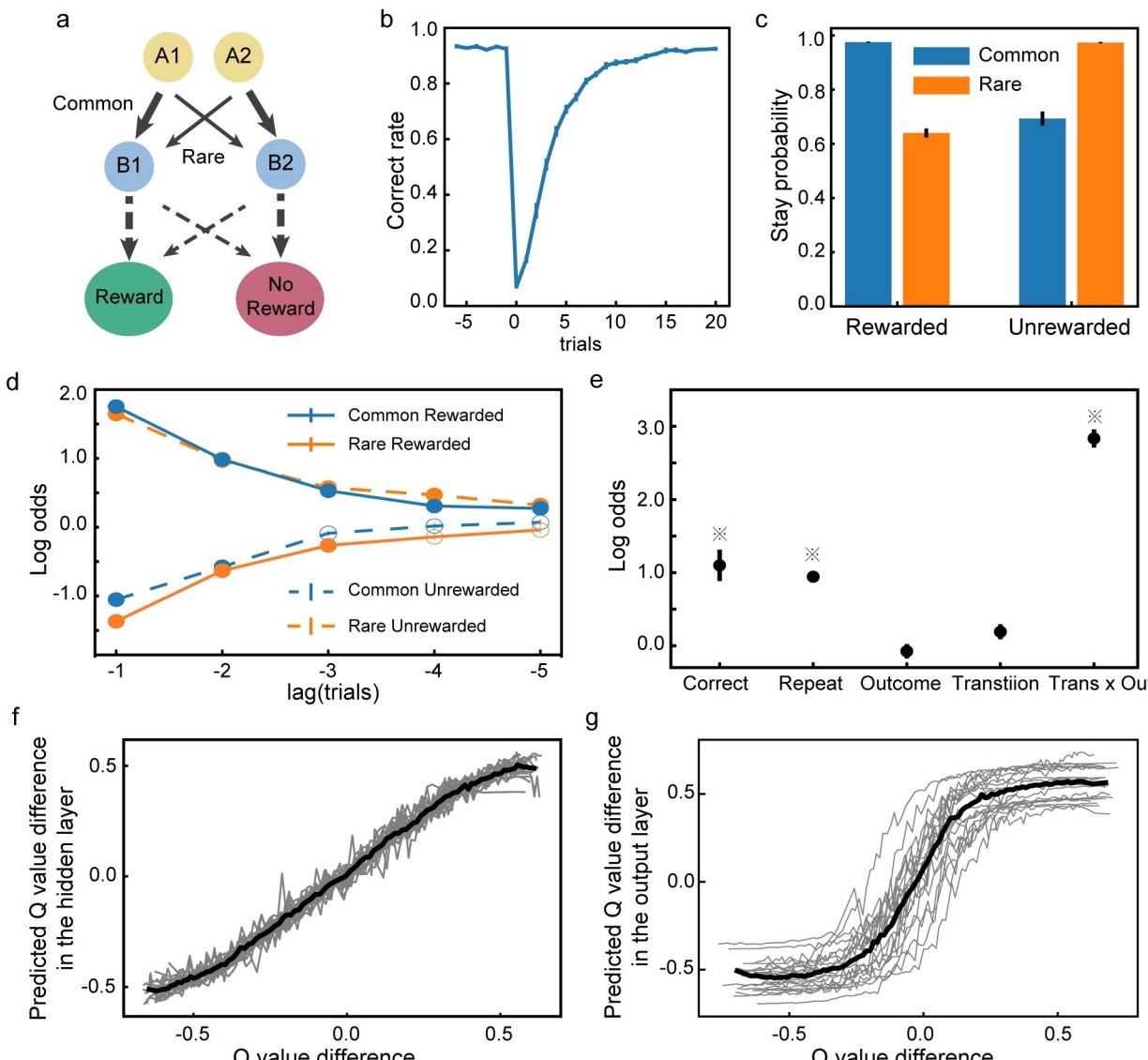

**Fig 10. Task 4: Two-step decision task. a**. The two-step task. The thick and thin lines denote the common and rare transitions, respectively. The contingencies indicated by the dashed lines are reversed across blocks. **b**. The switching behavior. Trials are aligned to the block switch (trial 0). The performance first drops to below the chance level but then gradually recovers. **c**. The probability of repeating the previous choice. The stay probabilities of the subsequent trials are higher for the CR and the RU trials than the RR and the CU trials. **d**. Trial history effects. The choice in the current trial is affected by the trial types in the previous trials. Solid dots indicate significant effect (Bonferroni correction, $p < 0.05$). **e**. Factors affecting the choices. ※ indicates significance ($p<0.01$). **f**. Units in the hidden layer encode the difference between the estimated Q-values of the two actions. Greys lines represent the predictions based on the units' activities in each run, and the black line is the average across runs. **g**. The response difference between the two choice output units is correlated with the difference between the estimated Q-values of the two actions.

unrewarded (RU). Then we calculate the probability of the network repeating the previous choice in the next trial. The probability is higher after the CR and RU trials than that after the CU and RR conditions (Fig 10C). This result indicates that the model exhibits a robust model-based behavior tendency, even more so than the human subject [27]. Interestingly, even though the error signals are only propagated to one trial back during the training, the history

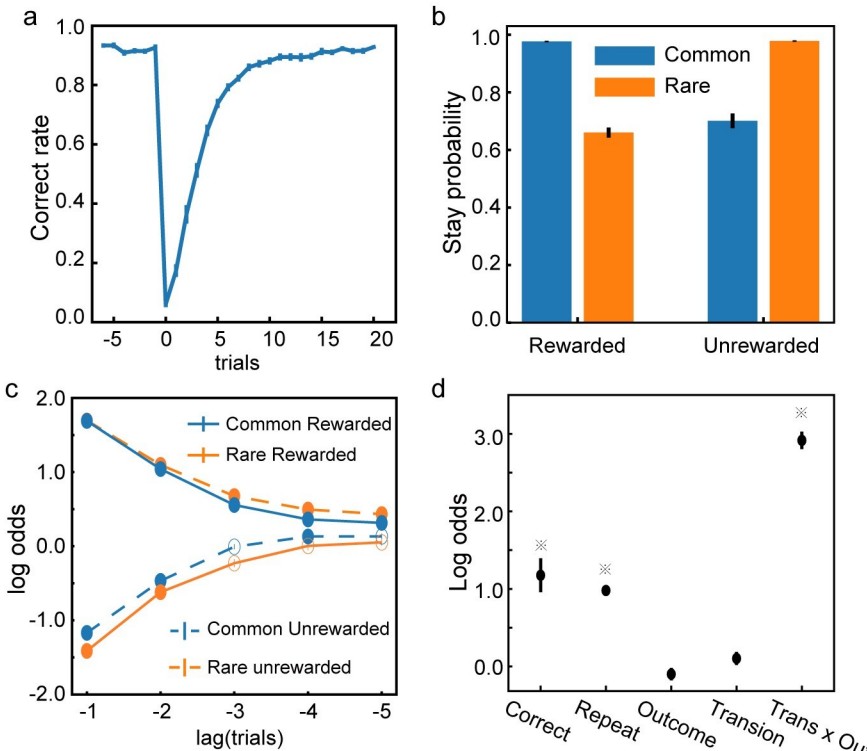

**Fig 11. Task 4: Model testing with different block size.** Same conventions as in Fig 10. Models are trained with 50-trial blocks but tested with 70-trial blocks. **a**. The switching behavior. Trials are aligned to the block switch (trial 0). **b**. The probability of repeating the previous choice. **c**. Trial history effects. Solid dots indicate significant effect (Bonferroni correction, p < 0.05). **d**. Factors affecting the choices. ※ indicates significance (p<0.01).

influence on the network's choice extends to many trials back, which is demonstrated by a logistic regression model with L1 penalty (Fig 10D).

Two more analyses corroborate this conclusion. We first use a logistic regression model to look at how a variety of factors affect the network's choices (Fig 10E). The result suggests that the interaction term between the intermediate outcome and the reward outcome (*Trans x Out*) is the factor most affecting the choices. We further fit the network's behavior with a mixture of a model-free and a model-based algorithm [27,28]. The higher weight for the model-based strategy ($w_{model\_based}$ = 0.87±0.04, $w_{model\_free}$ = 0.13±0.04) confirms that the network's choice behavior is closer to model-based.

Although the network is not trained explicitly to calculate and compare the value associated with each action, it is curious whether the network units encode the Q-value estimated with RL. Based on the RL model fitting above, we estimate the choices' Q-value in each trial. Using linear regression, we can decode the fitted Q-value difference between the two alternatives based on the unit activities in the hidden layer with high accuracy. (Fig 10F, explained variance:93.89±0.49%). The difference between the two output units' responses also depends on the Q-value difference between the two actions (Fig 10G). It can be better fitted with a steep sigmoid function than with a linear function ($\Delta$AIC = -2256.02 ± 158.60 and $\Delta$BIC = -2243.07 ± 158.60, sigmoid vs. linear). This reflects the binary action outcomes of the task. Although it has been common to use the findings of neurons in the brain that encode Q-value to support the idea that the brain implements RL [29], our results suggest an alternative explanation for these findings.

## Discussion

Here, we present a neural network framework inspired by the statistical inference approach of NLP for flexible and adaptive decision making. Similar to the neural network models used in NLP that can capture syntax structure from text sequences, our models learn the event contingencies from event sequences and use them to solve a range of decision-making and learning tasks. Importantly, we find that the model units exhibit characteristic response patterns of the decision-related neurons previously described experimentally, even though we do not train the network models to do so. These results suggest that our neural network framework based on statistical sequence learning may help us understand the neural computation and the neural circuitry underlying many decision-making and learning behaviors.

### Biological relevance

We have shown that the units in our model show response patterns similar to those of the neurons in a wide range of brain areas that have been implicated in decision making and learning [14,16,19,30]. These areas, including the prefrontal cortex, the posterior parietal cortex, and the basal ganglia, have broad connections with the sensory and motor cortical areas. They possess the necessary information to carry out the computations demonstrated here. At a local circuitry level, excitatory and inhibitory neurons in the cortex may together form a structure similar to a GRU and carry out computations in a similar fashion [9]. Therefore, the GRU network is likely to reflect a general computation principle in the brain.

In particular, we believe that our framework provides valuable insights into the understanding of the anatomy and functions of the basal ganglia. It has been long noticed that the key feature of GRU networks, which is the gating mechanism that controls the information flow, resembles the anatomical circuitry in the basal ganglia [8,31]. The basal ganglia receive broad inputs from all over the cortex, including the sensory cortices and motor cortices [32], which comprises all the necessary information for learning sequences.

Our framework may account for much of the current basal ganglia literature in several distinct directions, which have been largely studied separately. First of all, our modeling results explain the findings of the basal ganglia's role in decision making. For example, Ding and Gold found that the caudate neurons represented accumulated information in a random dot motion discrimination task [19,24]. Cisek and colleagues observed a group of neurons in the globus pallidus showed activity patterns similar to the urgency signal [20]. In rodent experiments, it has been reported that lesions in the striatum led to deficits in evidence integration and produced a bias in decision making [30,33]. Our framework may account for the results of these studies. Second, our framework, which is based on sequence learning, naturally explains the basal ganglia's role in performing action sequences and in procedure memory [34–36]. Third, the basal ganglia have also been indicated to mediate habitual and goal-directed behavior [37,38]; both are highly relevant to our framework. Our framework can be regarded as a model for habitual learning. Once trained, a model simply follows a set sequence of events and generate actions according to the acquired statistical structure, even in cases with a complex statistical structure. On the other hand, our framework may contribute to the understanding of the neural computation underlying goal-directed behavior. An essential component of goal-directed behavior is the ability to assess the consequences of actions. Such assessment depends on making predictions of future events, which is exactly what our framework does.

Our framework may, therefore, help us to link the existing studies of basal ganglia from distinct perspectives together to form a unified computational theory of basal ganglia's function. Future work may incorporate more details of basal ganglia circuitry for further investigations.

## Reinforcement learning

By learning the statistical relationships between sensory, action, and reward events across trials, our framework supports adaptive behavior. The reinforcement learning framework has been widely used to account for aniamls' adaptive behavior. However, our framework suggests a very different underlying mechanism from RL. During the training phase, our model learns the mapping between the states comprising past trial sequences, including reward events, and the corresponding actions. Afterwards during the testing phase, the apparent adaptive behavior of the model depends only on this mapping, but the internal network connections are fixed. No learning in terms of adjusting connection weights occurs. In contrast, reinforcement learning is believed to be implemented biologically by a brain circuitry that maintains and updates the representation of state values using reward feedbacks via synaptic plasticity [39].

Similar ideas have been explored in other recurrent network models [40,41]. Although many differences exist between the current framework and the previous ones, these networks depend on their internal dynamics to maintain a representation of the past reward and choice history. They most likely cannot replace reinforcement learning completely, especially for learning at large time scales, but they may provide a better account for the learning that happens in the brain at short time scales. Interestingly, units in our network also encode the Q-values estimated with reinforcement learning, making it difficult to distinguish the two scenarios with simple measures based on spiking activity. Experiments that directly manipulate synaptic plasticity or network dynamics may be used to test whether a particular adaptive behavior depends on network dynamics or synaptic plasticity.

## Training

To train the models, we need to feed them with appropriate sequences of sensory, action, and reward events. In the real brain, this means there has to be a separate mechanism to generate appropriate responses at appropriate moments and form these sequences in the first place to train the network. One solution is to start from the responses based on animals' innate responses or other established stimulus-action associations relevant to the task. These responses should contain a certain degree of variations that lead to different responses and reward outcomes [42]. For example, when we train monkeys to perform complicated behavior tasks such as the probabilistic reasoning task, we start from the most basic delayed saccade task. Monkeys have an innate tendency to make saccades toward newly appearing visual stimuli at variable delays. By selectively rewarding saccades with longer delays, we can train the monkeys to hold the fixation for an extended period before the saccade. More components of the task can be introduced to the task gradually this way. A similar strategy can be used to train our models. Starting with simple tasks that the network is capable of performing, even if only occasionally, we can selectively reinforce the behavior that leads to rewards and use the event sequences leading to rewards for the next stage training. Thus, the network can generate its own training data set with the rewards serving as the filters for the training sequences. Once the network performs at a satisfactory level, we may introduce more components and again depend on the operant variability to generate new sequences for further training. These procedures may be repeated until we reach the final version of the task. This training strategy allows us to train a naïve model in a manner similar to how experimenters train animals, and is conceptually similar to the curriculum training in the machine learning.

The training of our model is based on a supervised signal. The loss function, which is calculated between the predicted and the actual trial sequences and can be considered as a generalized prediction error signal, could come from the dopamine system in the brain. Midbrain dopamine neurons have been indicated to signal reward prediction error [43]. Yet, recent

findings have started to reveal a broader role of the dopamine neurons in signaling not just reward prediction error, but also predictions of sensory stimuli or actions [32,34,35,44]. These findings fit well with the current framework. Future experiments may reveal whether the dopamine system indeed fit this role.

### Experimental testing

Many of the predictions based on our framework can be tested experimentally. For example, we have discussed the *when/which* units in Task 1. One can directly record neurons from a brain area related to evidence accumulation, e.g. the basal ganglia, to see whether they may be sorted into two similar categories and contribute to decision making in a manner consistent with our model. Furthermore, one can look at such brain areas to see whether there is a representation of combined predictions of sensory, motor, and reward events during decision making as the framework indicates. The gating mechanism in our framework may have its correspondence in these brain areas, e.g., the direct and indirect pathways in the basal ganglia. One can test whether the model could predict the consequences of experimental manipulations of the gating mechanism. The difference between our model and standard RL models is also interesting. For example, in the two-step task, our model's performance depends on the network's ability to maintain history information across trials through its response dynamics. Therefore, we predict that disturbance during inter-trial intervals may disrupt the performance. This should not be expected if the brain uses a more RL-like strategy, in which the learning is based on synaptic plasticity instead of network dynamics.

### Further investigations

The current results focus on the testing of the most basic form of the GRU network. It would be interesting to explore in the future how other GRU network variants would provide a better account for the empirical data. In addition, GRU networks are able to solve problems at much larger scales (e.g., Chung et al., 2014). By expanding the investigation to more sophisticated networks, we might start to model more complex brain functions.

## Methods

### Task 1: Probabilistic reasoning task

**Network.**    Our framework contains three layers: an input layer, a hidden layer based on gated recurrent units, and an output layer (Fig 2A). The input layer contains units that carry information about the sensory, action, and reward events. ($N_{IL}$ = 20). There are 14 sensory input units. 10 of them represent the 10 shapes in the task. 3 additional units indicate the presence of the eye movement targets, including the fixation point, the left target, and the right target. We also include a unit that indicates the absence of any visual stimuli. There are 4 action input units that represent the efference copies of the motor commands: 1 for fixating on the fixation point, 1 for saccading to the left target, 1 for saccading to the right target, and 1 for saccading to other locations, which is considered as a fixation break and aborts a trial. Finally, 2 reward input units are included: one for the reward event and one for the absence of a reward. The output layer includes 20 units that mirror the inputs.

We run the network simulation with 100ms time steps in both the training and the testing. Our analyses and conclusions remain valid for smaller time steps.

**Behavior analysis.**    The network parameters are fixed during testing. We generate 5000 random shape sequences of 25 shapes as the testing data and feed them into the network model. A shape sequence is stopped whenever the output units associated with the saccades

are triggered. If the network does not make a response before all 25 shapes have been presented or makes a response at an inappropriate time point, the trial is aborted ($172.50 \pm 3.73$ or $3.45 \pm 0.07\%$ trials in each run, excluded in further analyses).

**Subjective weights (Fig 3B).**   We perform logistic regression to assess how each shape affects the choice. The probability of choice is a function of the sum of leverages, $Q$, provided by the shapes:

$$P(choice = left) = \frac{1}{1 + 10^{-Q}}, \tag{15}$$

$$Q = \beta_0 + \sum_{m=1}^{10} \beta_m N_m, \tag{16}$$

where $N_m$ represents how many times shape $m$ appears in a trial. $\beta_0$ is the bias term, $\beta_{1\sim10}$ are the estimates of how much weight the network model assigns to shape types 1 to 10. They are termed as the subjective weights of evidence [18]. Since the regressors are not independent of each other, we use ridge regression to minimize the variation of estimations. The hyperparameter controls the tradeoff between the cross-entropy loss, and the L2-norm of the coefficients is selected through ten-fold cross-validation.

**Shape order (Fig 3D).**   To test how the shape order affects the network's choice, we perform logistic regression on the trials with more than six epochs:

$$Q = \beta_0 + \sum_{n=1}^{3} \beta_n \text{logLR}_n + \beta_4 \sum_{n=3}^{N-3} \text{logLR}_n + \sum_{n=1}^{2} \beta_{n+4} \text{logLR}_{N-n}, \tag{17}$$

where $\text{logLR}_n$ is the logLR of the shape in epoch $n$ and $N$ is the total number of epochs in a sequence. $\beta_0$ is the bias term, $\beta_1$, $\beta_2$, $\beta_3$ the fitting coefficients of the shapes in the first three epochs, $\beta_4$ the average effect of the shapes in the middle epochs, and $\beta_5$, $\beta_6$ the second and the third epochs to the last. The regression is performed without the final shape ($n$-th). This is because the last shape is almost always (>99% of trials) consistent with the choice. The fitting procedure is similar to what we use above to estimate the subjective weights.

**Unit selectivity (Fig 5A and 5B).**   We test whether each neuron's activity in the hidden layer is modulated by the total logLR, absolute value of the total logLR, the urgency, and the choice outcome with linear regression. We align the hidden unit state $h$ to the shape onset in each epoch and use linear regression to characterize the unit's selectivity:

$$h_t = \beta_0 + \beta_1 \sum_{n=1}^{t} \text{logLR}_n + \beta_2 \left| \sum_{n=1}^{t} \text{logLR}_n \right| + \beta_3 u_t + \beta_4 c, \tag{18}$$

where $h_t$ is the unit's response at time $t$ aligned to the shape onset, $\sum_{n=1}^{t} \text{logLR}_n$ is the total logLR of the shapes that have been presented by time $t$, $u_t$ represents the urgency, which is quantified as the number of shape epochs, $c$ represents the choice of the network, which is set to 1 when the left target is chosen, -1 when the right target is chosen, and 0 if fixation is still maintained by the end of the epoch. The regression is performed at every time step during the shape representation and the inter-shape interval periods. We define a unit as selective to a variable if the variable shows significant effects on the unit's activity at every time step in an epoch. The significance is determined by two-tailed t-tests with Bonferroni correction.

**Response variability (Fig 5D).**   We calculate a unit's response variability as the standard deviation of the unit's average response in each shape epoch across trials. Fig 5D plots the average response variability for all units that are selective to the total logLR in all 20 simulation runs, using only the trials with more than 5 shapes.

**Speed-accuracy tradeoff (Fig 6D).**   To test how *when* units affect the speed-accuracy tradeoff, we suppress the outputs of the +*when*/-*when* units that are selected with variable criteria. For example, the criteria of the accumulative $I_{when}$ of the selected +*when* units are set to

10%, 20%, 30%, 40%, and 50% out of the summed $I_{when}$ of all units with positive $I_{when}$. At each criterium, the number of units manipulated is not linearly scaled, but their total contribution to the behavior, measured by their summed $I_{when}$, is scaled linearly.

**Network analysis (Fig 6E).** Geodesic distance and maximum flow are used to quantify how much information may be transferred between two nodes in a graph. We use the weight matrix $U_h$, which represents the connection strengths between the units in the hidden layer, to construct a weighted directed graph. Each unit in the hidden layer corresponds to a node in the graph. Connections with the highest 30% of the absolute values of the connection strength are turned into the edges in the graph. The results hold if we keep the highest 10% or 50% of the connections. The weight and capacity of each edge are defined as the absolute value of the original connection strength.

We define the geodesic distance from node A to target B as the minimum summed inverse of the weights of the edges from node A to B [45]. The geodesic distances between any pairs are calculated with Dijkstra's algorithm [46]. The distance between unit pairs not connected is defined as infinite. To account for these pairs, we compare the mean of the inverse of the geodesic distance across all unit pairs in each group.

## Task 2: Multisensory integration task

**Input units.** There are two groups of input units representing the sensory stimuli, corresponding to the visual and the vestibular inputs. Each group has 8 units with Gaussian tuning curves with different preferred directions $D_{pref}$ at -90˚, -64.3˚, -38.6˚, -12.9˚, 12.9˚, 38.6˚, 64.3˚, and 90˚, respectively. The tuning curve $f(D)$ is described as the following:

$$f(D) = N(D - D_{pref}, \sigma^2), \tag{19}$$

where $D$ is the direction of the motion stimulus (-90˚ $\leq D \leq$ 90˚), and $\sigma = 45$˚ is the standard deviation of the Gaussian function. The observed spike count $s$ has a Poisson distribution:

$$p(s|D) = \frac{e^{-gf(D)}(g \cdot f(D))^s}{s!}, \tag{20}$$

where $g = 300$ is a constant for the gain. Finally, to limit the response of the input units to be smaller than 1, a sigmoid-like function is used to calculate the response $r(s)$:

$$r(s) = 1/(1 + e^{1-s}) \tag{21}$$

The exact choice of the function is not essential. Under the bimodal condition, the responses of both the visual and the vestibular input units are calculated with the heading direction $D$. Under the unimodal condition, the responses of the units of the unavailable modality are set to 0.5 (neutral) without noise.

**Bayesian inference.** We estimate the heading direction by calculating the discretized posterior probability of the possible motion directions;

$$p(D|s) \propto e^{\sum_i (-g \cdot f_i(D) + s_i \cdot \log(g \cdot f_i(D)))} \tag{22}$$

The left choice target is chosen if the probability of motion direction towards left, $p(D<0|s)$, is larger than the probability of motion direction towards right, $p(D>0|s)$. Otherwise, the right target is chosen.

In the bimodal condition, we integrate the information from the visual and the vestibular inputs and calculate the posterior probability $p(D|s_{vis}, s_{vest})$, which is proportional to the product of the $p(D|s_{vis})$ and $p(D|s_{vest})$, given the independence between $s_{vis}$ and $s_{vest}$.

All analyses are based on 20 simulation runs.

## Task 3: Confidence / Post-decision wagering task

**Input units.** The network has 22 input units. Ten input units are visual units that respond to motion stimuli. Five of them prefer the leftward motion, and the other five prefer the rightward motion. Their internal activation, $s$, is linearly scaled with the coherence. An independent Gaussian noise, $N(0,\sigma^2)$, is added to mimic the noise during the sensory processing, where $\sigma = 2.5$. A sigmoid function is used as the activation function.

$$s = coh\% \cdot 10 + N(0, \sigma^2) \tag{23}$$

$$r = sigmoid(s) \tag{24}$$

where the coherence of the moving dots, *coh*, is positive when the motion direction matches the unit's preferred direction and negative when not.

The momentary evidence $e_t$ is the response difference between the leftward-preferring units and the rightward-preferring units. The accumulated evidence $E$ is then the sum of momentary evidence across time:

$$e_t = \sum_i s_{left}^{t,i} - \sum_i s_{right}^{t,i}, \tag{25}$$

$$E = \sum_{t=1}^{T} e_t, \tag{26}$$

where $s_{left}^{t,i}$ and $s_{right}^{t,i}$ are the internal activities at time point $t$ of the $i$-th unit preferring the leftward and the rightward motion, respectively. $T$ is the duration of the motion stimulus and is randomly selected from [1, 2, 3, 4, 6, 8] at equal probabilities. The final decision is based on the accumulated evidence $E$. The trial sequences used to train the model is generated as follows [26]. When the sure target is not available, the left target is chosen if $E$ is larger than 0. The right target is chosen if $E$ is smaller than 0. When the sure target is given, the left and right choice targets are selected only if $|E|$ is larger than a pre-defined threshold $\theta$. Otherwise, the sure target is selected. Furthermore, the threshold $\theta$ increases linearly over time:

$$\theta = 5 + 2.5t \tag{27}$$

20 independent runs are simulated for testing the reproducibility.

**Unit activity analysis.** The analysis is based on the choice-selective units. These units are chosen based on their activities at the time point right before the choice in the trials without the sure target. The selectivity is determined by the two-tailed t-test with Bonferroni correction. The choice direction that induces a larger response is defined as a unit's preferred direction (left: n = 46.75 ± 1.93, right: n = 42.45 ± 1.85).

## Task 4: Two-step Task

**Simulation.** We use 20 trained sessions in our analysis. Each session contains $7.5^*10^5$ training trials (15000 blocks of 50 trials) and 100 testing blocks.

**Model fitting.** The analysis was previously described [27,28,40]. Briefly, the model choice is fitted to a mixed model-free and model-based algorithm. For the model-free algorithm, the value of the chosen options and the observed intermediate outcomes are updated by a temporal difference algorithm with two free parameters: the learning rate $\alpha_1$ and the eligibility $\lambda$, representing the proportion of the second stage reward prediction error that is attributed to the first-stage chosen options A1 and A2. In the model-based algorithm, the value of the observed intermediate outcome is also learned with a temporal difference algorithm. The value of each option is the sum of the products of the transition probabilities and the values of two

intermediate outcomes. An extra parameter, learning rate $\alpha_2$, is used for the update of the value of the intermediate outcomes in the model-based algorithm. The overall value of each option is the weighted average of its values calculated with the model-free and model-based algorithm ($w_{model\text{-}free} + w_{model\text{-}based} = 1$). Finally, the probability of choosing each option is a softmax function of the values of the two options:

$$p(action = A_i) = \frac{exp(\beta \cdot v(A_i) + p \cdot rep(A_i))}{\sum_{A\prime} exp(\beta \cdot v(A\prime) + p \cdot rep(A\prime))} \tag{28}$$

where inverse temperature parameter, $\beta$, controls the randomness of choice, $rep(A_i)$ is set to 1 if action $A_i$ is chosen in the previous trial, and parameter $p$ captures the tendency of repeating the previous trial, and $v(A_i)$ is the value of option $A_i$. Together, there are six free parameters, and a maximum likelihood estimation algorithm is used for fitting.

**Logistic regression.**   The analysis was described previously [28,40]. Briefly, five potential factors are tested with logistic regression. They are *Correct*—a tendency to choose the choice with higher reward probability; *Repeat*—a tendency to repeat the choice no matter what the reward outcome is; *Outcome*—a tendency to repeat the rewarded choice in the previous trial; *Transition*—a tendency to repeat the choice when it leads to the common intermediate outcome and switch when it leads to the rare intermediate outcome; *Trans x Out*–a tendency to repeat the same choice when the previous trial is CR or RU, and to switch the choice if the previous trial is CU or RR.

## Acknowledgments

We thank Zhongqiao Lin, Chechang Nie, Yang Xie, Wenyi Zhang for their help in all phases of the study, and Shan Yu for providing comments and advice.

## Author Contributions

**Conceptualization:** Tianming Yang.

**Data curation:** Zhewei Zhang, Tianming Yang.

**Formal analysis:** Zhewei Zhang, Tianming Yang.

**Funding acquisition:** Tianming Yang.

**Investigation:** Zhewei Zhang, Huzi Cheng, Tianming Yang.

**Methodology:** Zhewei Zhang, Huzi Cheng, Tianming Yang.

**Project administration:** Tianming Yang.

**Supervision:** Tianming Yang.

**Validation:** Zhewei Zhang, Tianming Yang.

**Visualization:** Zhewei Zhang, Tianming Yang.

**Writing – original draft:** Zhewei Zhang, Tianming Yang.

**Writing – review & editing:** Zhewei Zhang, Tianming Yang.

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
