## [Decision Letter · Decision Letter 0]

17 Jun 2020

Dear Dr. Yang,

Thank you very much for submitting your manuscript "A Recurrent Neural Network Framework for Flexible and Adaptive Decision Making based on Sequence Learning" for consideration at PLOS Computational Biology.

As with all papers reviewed by the journal, your manuscript was reviewed by members of the editorial board and by several independent reviewers. In light of the reviews (below this email), we would like to invite the resubmission of a significantly-revised version that takes into account the reviewers' comments.

We cannot make any decision about publication until we have seen the revised manuscript and your response to the reviewers' comments. Your revised manuscript is also likely to be sent to reviewers for further evaluation.

Sincerely,

Alireza Soltani

Associate Editor

PLOS Computational Biology

Samuel Gershman

Deputy Editor

PLOS Computational Biology

Reviewer's Responses to Questions

**Comments to the Authors:**

Reviewer #1: This manuscript describes a RNN based on GRU units, which is trained to solve a fairly complex sequential probabilistic task. Specifically a sequence composed of an alphabet of 10 symbols (each of which is probabilistically associated with a correct Left/Right response) is presented until a choice is made. The optimal strategy is to integrate information across symbols until enough evidence for a decision is available. As the authors point out this task has similarities with natural language processing, thus the use of a GRU. After training the RNN solves the tasks across nontrained sequences. The units of the RNNs seem to qualitatively resemble the behavior of neurons recorded during animal experiments using similar tasks, including response, log likelihood and “urgency” neurons. The analyses and simulations seem well executed, however a weakness is the conclusions in terms of how the results improve our understanding of brain function. Some reference is made regarding capturing the properties of basal ganglia, but there is no direct clear link.

Can the authors make any clear cut experimental predictions? For example are the authors predicting that animals are attempting to predict both the subsequent stimuli and reward with equal weighting (as implemented by their loss function)?

The authors attempt to make a link to the biology, and make a few general statements about similarities to the basal ganglia, and a link between GRUs and the basal ganglia circuitry. This seems counterintuitive to me for a number of reasons including the fact that the basal ganglia circuitry is all inhibitory. In regards to the potential biological relevance it would be important to determine if gated units (which can have infinite long memory) are needed. Indeed, since no parametric studies are presented it would be helpful to do so, and one way this could be accomplished is to compare GRU performance with ReLU performance. That is, is this a task that requires the essentially infinite long-term memory of gated units, or can it be achieved equally well by more realistic ReLUs?

Does performance exhibit a commutative property. That is, is the Reaction time the same for shapes 1,2,3 and 3,2,1 (controlling for cases in one shape basically solves the task)? This is an important question because from a mathematical stance it should be the case, but biologically speaking it is probably not the case, because animals are generally heavily biased by recency effects. Relatedly is the model equally weighing all evidence or preferentially weighing more recent evidence (i.e., forgetting early evidence)?

Something that is a bit glossed over in the presentation is that the network is not actually making a decision, there seems to be a postprocessing stage in which information from the network is analyzed to see if a criteria is reached and then the simulation is terminated. This is a bit misleading as many people will be left with the impression the RNN is autonomously making a reaction time decision. Thus, this postprocessing stage should be strongly emphasized in the results.

Apparently, the model was trained with Adam, but absolutely no information is given about this critical component of any model.

It would be helpful to show some raw data of model performance as is often shown for standard RNN models during the performance of the task.

The performance measure does not allow the reader to understand how well the model is integrating, how well can it perform just looking at the last N shapes?

Line 42. “framework”.

Line 92. “ranging”

Li 280. Just “Performance” (behavior gives the impression there were animal studies).

Li 735 “reported THAT lesions”

Reviewer #2: In the manuscript titled “A Recurrent Neural Network Framework for Flexible and Adaptive Decision Making based on Sequence Learning”, Zhang and Colleagues used natural language processing (NLP) framework and trained a network of gated recurrent units in four different experimental setting. Specifically, similar to networks in the NLP that are trained to predict text sequences, authors trained their networks to take inputs in the form of event sequences (sensory and reward outcome events) and predict future events through supervised learning. Networks were trained on ¬¬a probabilistic reasoning task, a multi-sensory integration task, a confidence/post-decision wagering task, and a two-step task. Authors showed that networks can learn perform the task and showed behavior similar to that observed to animals. Additionally, authors found units that resembled activity of recorded neurons in different areas of the brain.

Overall, I found the paper suitable for publication in journal of PCB and their proposed framework interesting. However, some details of training procedure and analysis require further justification. Moreover, strong conclusions are drawn without enough supporting information, which needs to be addressed. Please find my comments below:

Major concerns:

1) Page 14, lines 267-270: Why a drift-diffusion model (DDM) with collapsing boundary was used to simulate the choice used for training the network? It is sounds circular to train the network with a certain model and observe units that resemble the key parts of that model. How much of these observations depend on the model behind choice behavior? Why not train the network with the optimal algorithm? Authors need to clarify this.

2) Page 20, lines 408-409: Authors’ definition of Iwhen and Iwhich are very confusing to me. These measures don’t seem to be doing what they are supposed to do. For example, if a unit has (WLT, WRT, WFT) respectively equal to (5, 2, 3.5), this unit is +which (5-2=3) but apparently not sensitive to fixation at all ((5+2)/2-3.5=0), which does not make sense. Why not pass them through a psychometric function and use their relative weights to compare units?

3) Page 22, Figure 6: Can authors explain why the connection weights for +/- when and +/- which are not symmetric with respect to 0. Wouldn’t it show that not only weights, but the activity of these units should be also considered?

4) Page 30, lines 624-627: I don’t understand why only rewarded trails are used to train the network. What happens if all trials are fed into the network during the training? Unrewarded sequences carry same amount of information about the task as the rewarded trials.

5) Page 32, line 671-673, & Page 37-39: Even though that the loss function of the trained network is not that of reinforcement learning, the network has access to the reward, its estimate of reward at each time point, and is required to match its actions to rewarding ones (necessary pieces for a reinforcement learning agent). So, what makes authors think that the network is not using some deviation of RL to solve the task? Recent studies have been focused on the relation of these two frameworks [1-2]. Authors cannot make such a strong claim considering the structure of their network.

6) In all these experiments, only a single network is trained and used to calculate the average and SE values. How consistent are these results for different trained networks? What percentage of the trained networks show the mentioned behaviors?

Minor concerns:

1) Page 3, line 48-49: There seems to be a few words missing/incorrect in this sentence;

“Therefore, we build a recurrent neural network framework based on the gated recurrent units and test how it matches experimental findings in four exemplar tasks, each focusing on a different aspect of decision making and learning.”

2) Page 7, last paragraph: Please add a more detailed summary of your results in this section. The language is very vague for the introduction.

3) Page 9, line 147: “Our network framework contains three layers … ”

4) I did not find any information on the criteria for training procedure. Was the training stopped after the error went below a certain threshold? If so, please add it to the related section.

5) Please add a more detailed caption to Figures 4 and 11.

6) Page 21, line 427: “The where and when group units only overlap rarely … ”

7) Page 21, line 429: Please report the prevalence of each group separately.

8) Page 22, Figure 6a: Please add a horizontal line for 0 to both panels.

9) Page 26, Figure 8b: Please add information on the error bars to the caption.

10) Page 28, line 587: Please report the prevalence of these units in your model.

References:

[1]. Luketina, J., Nardelli, N., Farquhar, G., Foerster, J., Andreas, J., Grefenstette, E., ... & Rocktäschel, T. (2019). A survey of reinforcement learning informed by natural language. arXiv preprint arXiv:1906.03926.

[2]. Jiang, Y., Gu, S. S., Murphy, K. P., & Finn, C. (2019). Language as an abstraction for hierarchical deep reinforcement learning. In Advances in Neural Information Processing Systems (pp. 9414-9426).

**Have all data underlying the figures and results presented in the manuscript been provided?**

Reviewer #1: Yes

Reviewer #2: None

PLOS authors have the option to publish the peer review history of their article (what does this mean?). If published, this will include your full peer review and any attached files.

Reviewer #1: No

Reviewer #2: No
---

## [Decision Letter · Decision Letter 1]

31 Aug 2020

Dear Dr. Yang,

Thank you very much for submitting your manuscript "A Recurrent Neural Network Framework for Flexible and Adaptive Decision Making based on Sequence Learning" for consideration at PLOS Computational Biology. As with all papers reviewed by the journal, your manuscript was reviewed by members of the editorial board and by several independent reviewers. The reviewers appreciated the attention to an important topic. Based on the reviews, we are likely to accept this manuscript for publication, providing that you modify the manuscript according to the review recommendations.

**Importantly, please add a few sentences explaining the results of analyses in response to Reviewer # 2. Please note that we will not send the manuscript to review again and the final decision will be made at the editorial level. In addition, in accordance with the journal policy, please also make your data and/or codes available if you have not done already. **

Sincerely,

Alireza Soltani

Associate Editor

PLOS Computational Biology

Samuel Gershman

Deputy Editor

PLOS Computational Biology

[LINK]

Reviewer's Responses to Questions

**Comments to the Authors:**

Reviewer #1: The authors have done a reasonable job in addressing my concerns.

Fig 1a should read “Pavlovian”

Reviewer #2: I would like to thank the authors for responding to my questions. I have a recommendation to add:

RE 2.3 and 2.4: I suggest the authors to add a few sentences to the main manuscript explaining these points.

**Have all data underlying the figures and results presented in the manuscript been provided?**

Reviewer #1: Yes

Reviewer #2: Yes

PLOS authors have the option to publish the peer review history of their article (what does this mean?). If published, this will include your full peer review and any attached files.

Reviewer #1: No

Reviewer #2: No
---

## [Editor Report · Decision Letter 2]

16 Sep 2020

Dear Dr. Yang,

We are pleased to inform you that your manuscript 'A Recurrent Neural Network Framework for Flexible and Adaptive Decision Making based on Sequence Learning' has been provisionally accepted for publication in PLOS Computational Biology.

Best regards,

Alireza Soltani

Associate Editor

PLOS Computational Biology

Samuel Gershman

Deputy Editor

PLOS Computational Biology

---

## [Editor Report · Acceptance letter]

22 Oct 2020

PCOMPBIOL-D-20-00841R2 

A Recurrent Neural Network Framework for Flexible and Adaptive Decision Making based on Sequence Learning

Dear Dr Yang,

I am pleased to inform you that your manuscript has been formally accepted for publication in PLOS Computational Biology. Your manuscript is now with our production department and you will be notified of the publication date in due course.

With kind regards,

Kaitlin Butler
